# Suppression of Superficial Microglial Activation by Spinal Cord Stimulation Attenuates Neuropathic Pain Following Sciatic Nerve Injury in Rats

**DOI:** 10.3390/ijms21072390

**Published:** 2020-03-30

**Authors:** Masamichi Shinoda, Satoshi Fujita, Shiori Sugawara, Sayaka Asano, Ryo Koyama, Shintaro Fujiwara, Kumi Soma, Takaaki Tamagawa, Tomoyuki Matsui, Daisuke Ikutame, Masatoshi Ando, Ayaka Osada, Yuki Kimura, Kazutaka Kobayashi, Takamitsu Yamamoto, Kuniko Kusama-Eguchi, Masayuki Kobayashi, Yoshinori Hayashi, Koichi Iwata

**Affiliations:** 1Department of Physiology, Nihon University School of Dentistry, Tokyo 101-8310, Japan; hayashi.yoshinori@nihon-u.ac.jp (Y.H.); iwata.kouichi@nihon-u.ac.jp (K.I.); 2Department of Biology, Nihon University School of Dentistry, Tokyo 101-8310, Japan; fujita.satoshi@nihon-u.ac.jp; 3Department of Psychosomatic Dentistry, Graduate School of Medical and Dental Science, Tokyo Medical and Dental University, Tokyo 113-8510, Japan; 0629ompm@tmd.ac.jp; 4Department of Oral Diagnostic Sciences, Nihon University School of Dentistry, Tokyo 101-8310, Japan; bunny2838@gmail.com; 5Department of Oral and Maxillofacial Surgery, Nihon University School of Dentistry, Tokyo 101-8310, Japan; dery18014@g.nihon-u.ac.jp (R.K.); tamagawa.takaaki@nihon-u.ac.jp (T.T.); dema17002@g.nihon-u.ac.jp (M.A.); deyu18010@g.nihon-u.ac.jp (Y.K.); 6Department of Complete Denture Prosthodontics, Nihon University School of Dentistry, Tokyo 101-8310, Japan; desh19025@g.nihon-u.ac.jp (S.F.); deda17003@g.nihon-u.ac.jp (D.I.); 7Department of Pediatric Dentistry, Nihon University School of Dentistry, Tokyo 101-8310, Japan; deku16016@g.nihon-u.ac.jp (K.S.); deto17027@g.nihon-u.ac.jp (T.M.); 8Department of Orthodontics, Nihon University School of Dentistry, Tokyo 101-8310, Japan; deay18005@g.nihon-u.ac.jp; 9Division of Neurosurgery, Department of Neurological Surgery, Nihon University School of Medicine, Tokyo 173-8610, Japan; kobayashi.kazutaka@nihon-u.ac.jp (K.K.); yamamoto.takamitsu@nihon-u.ac.jp (T.Y.); 10Laboratory of Biochemistry, Nihon University School of Pharmacy, Chiba 274-8555, Japan; kusama.kuniko@nihon-u.ac.jp; 11Department of Pharmacology, Nihon University School of Dentistry, Tokyo 101-8310, Japan; kobayashi.masayuki@nhon-u.ac.jp

**Keywords:** spinal cord stimulation, spared nerve injury, microglial activation, somatosensory cortex, in vivo optical imaging

## Abstract

We evaluated the mechanisms underlying the spinal cord stimulation (SCS)-induced analgesic effect on neuropathic pain following spared nerve injury (SNI). On day 3 after SNI, SCS was performed for 6 h by using electrodes paraspinally placed on the L4-S1 spinal cord. The effects of SCS and intraperitoneal minocycline administration on plantar mechanical sensitivity, microglial activation, and neuronal excitability in the L4 dorsal horn were assessed on day 3 after SNI. The somatosensory cortical responses to electrical stimulation of the hind paw on day 3 following SNI were examined by using in vivo optical imaging with a voltage-sensitive dye. On day 3 after SNI, plantar mechanical hypersensitivity and enhanced microglial activation were suppressed by minocycline or SCS, and L4 dorsal horn nociceptive neuronal hyperexcitability was suppressed by SCS. In vivo optical imaging also revealed that electrical stimulation of the hind paw-activated areas in the somatosensory cortex was decreased by SCS. The present findings suggest that SCS could suppress plantar SNI-induced neuropathic pain via inhibition of microglial activation in the L4 dorsal horn, which is involved in spinal neuronal hyperexcitability. SCS is likely to be a potential alternative and complementary medicine therapy to alleviate neuropathic pain following nerve injury.

## 1. Introduction

Neuropathic pain such as postherpetic neuralgia, post-stroke pain, and trigeminal neuralgia is known to occur as a result of peripheral and/or central neurological disturbances [1]. Neuropathic pain is also clinically typified by intractable non-noxious stimulation-induced pain and thought to be a direct result of a lesion or disease that affects the peripheral somatosensory system [2]. In clinical practice, spinal cord stimulation (SCS) has been well established as a safe and effective treatment of chronic, intractable pain including neuropathic pain [3,4]. Sciatic nerve injuries have been widely employed as an animal model of neuropathic pain, and SCS reportedly improves the intractable neuropathic pain following nerve injuries [5,6,7,8,9]. Presently, the prime analgesic effects of SCS involves the activation of the descending pain inhibitory system [4,10].

Many studies have indicated that persistent mechanical hypersensitivity was induced in the hind paw following sciatic nerve injuries along with microglial cell accumulation in the spinal dorsal horn (SDH), and the suppression of microglial cell activation in the spinal cord significantly alleviated mechanical hypersensitivity [11,12,13]. In SDH, the increased Iba1 immunoreactivity that indicates the hyperactive states of microglia was reported to be detected as early as 20 min, peaking on day 3, and remaining at a significant level on day 7 following peripheral nerve injury [14].

Nevertheless, it is still quite uncertain as to whether SCS will suppress SDH microglial activation and neuronal hyperexcitability following sciatic nerve injury. To evaluate the mechanisms underlying the SCS-induced modulation of neuropathic pain associated with sciatic nerve injury, we examined the effect of SCS on mechanical hypersensitivity and dorsal horn microglial activation, and assessed neuronal hyperexcitability in the dorsal horn following sciatic nerve injury.

Moreover, it is well known that pathological spinal nociceptive information originating from peripheral nerve injury is conveyed to supraspinal components of the central nervous system (CNS) such as the primary and secondary somatosensory cortices and limbic cortices [15]. The rewiring of the CNS circuitries and alterations of various molecular generation mechanisms occur in the CNS nociceptive pathways following peripheral nerve injury. These neuroplastic changes in the CNS nociceptive pathways are thought to be involved in the occurrence of persistent pain associated with peripheral nerve injury [16]. Thus, it is essential to determine whether SCS affects the neuronal activities in these upper CNS areas to evaluate the mechanisms underlying SCS-induced analgesic action on the persistent pain associated with peripheral nerve injury. We, therefore, performed in vivo optical imaging with a voltage-sensitive dye in the somatosensory cortex to elucidate whether SCS changes the spatiotemporal profiles of cortical responses to electrical stimulation of the hind paw following sciatic nerve injury.

## 2. Results

### 2.1. SCS and Minocycline Depressed Plantar Mechanical Hypersensitivity Following SNI

Paw withdrawal thresholds (PWTs) were remarkably decreased on day 3 following spared nerve injury (SNI) compared to naive animals in both SCS and sham-stimulation animals (*p* < 0.001 vs. naive; Figure 1a). SCS for six hours successively significantly recovered the decrement of PWT at 30 min post-SCS (post-sham stimulation (*n* = 6), SCS (*n* = 7), *p* = 0.01). Moreover, PWT at 12 h after 6 h of successive SCS tended to be higher than that obtained in the post-sham stimulation group (*p* = 0.09).

Successive intraperitoneal minocycline administration kept suppressing the SNI-induced decrement in PWT on day 3 following SNI, compared with successive intraperitoneal vehicle (saline) administration (vehicle (*n* = 7), SCS (*n* = 7), *p* = 0.09) (Figure 1b).

### 2.2. Microglial and Astroglial Activation in the L4 Dorsal Horn

The relative amount of Iba1 protein increased in the L4 SDH on day 3 following SNI (*n* = 4) compared with the control (*n* = 4) (Figure 2). Iba1-immunoreactive (IR) cells were observed in laminae II of the dorsal horn in the L4 spinal cord bilaterally on day 3 after SNI and 12 h after either SCS or sham-stimulation (Figure 3a). The morphology analysis of microglia was performed as previously described [17]. In the post-sham stimulation group, Iba1-IR cells changed their morphological features such as large soma and short processes on day 3 after SNI (Figure 3a, an inlet on the left). The ipsilateral density of Iba1-IR cells with large soma and short processes on day 3 after SNI with post-sham stimulation increased, and the increment in the ipsilateral density of Iba1-IR cells was significantly suppressed by SCS stimulation (post-sham (*n* = 3): 12.8 ± 1.3; SCS (*n* = 3): 7.4 ± 0.3, *p* = 0.004) (Figure 3b). On the other hand, there was no significant difference in the contralateral density of Iba1-IR cells on day 3 after SNI with SCS or post-sham stimulation.

Moreover, the ipsilateral increased density of Iba1-IR cells on day 3 after SNI with post-sham stimulation tended to be suppressed by intraperitoneal (i.p.) minocycline administration (vehicle (saline) (*n* = 3): 9.9 ± 0.9; minocycline (*n* = 3): 6.0 ± 1.3, *p* = 0.07) (Figure 3c).

There was no significant difference in the ipsilateral and contralateral densities of glial fibrillary acidic protein (GFAP)-IR cells on day 3 after SNI with SCS or post-sham stimulation (Figure 4).

### 2.3. Effect of SCS on L4-S1 Nociceptive Neuronal Activity Following SNI

Wide dynamic range (WDR) neurons in laminae II of L4-5 SDH showed responses to non-noxious and noxious mechanical stimulation of the hind paw (post-sham stimulation: six WDR neurons from two rats; post-SCS stimulation: five WDR neurons from three rats, Figure 5a). The mean number of spikes to 26 g (post-sham stimulation: 17.3 ± 4.6; post-SCS stimulation: 18.2 ± 0.9, *p* = 0.25) and 60 g (post-sham stimulation: 28.6 ± 4.9; post-SCS stimulation: 48.2 ± 0.9, *p* = 0.006) mechanical stimuli were significantly smaller in SNI rats with post-SCS stimulation compared with those in post-sham stimulation.

Furthermore, the mean number of spikes to pinch stimuli (post-sham stimulation: 52.1 ± 7.0; post-SCS stimulation: 7.7 ± 1.3, *p* = 0.003) and brush stimuli (post-sham stimulation: 32.8 ± 8.9; post- SCS stimulation: 0.6 ± 0.1, *p* = 0.04) were significantly smaller in SNI rats with post-SCS stimulation than in those receiving post-sham stimulation (Figure 5b).

### 2.4. Cortical Responses to Electrical Stimulation of the Hind Paw

In optical imaging experiments, we used rats that received SNI. In agreement with a previous study [18], response to stimulation of the contralateral hind paw (Figure 6c,d) was initiated and spread over the surrounding region that was caudal and medial to the region initially activated by stimulation of the contralateral forepaw in a control rat (Figure 6a,b,g). In a rat that received SCS, stimulation of the contralateral hind paw similarly induced a cortical response, but the activated area was relatively small (Figure 6e,f). These results suggest that the pathway from the contralateral hind paw to the somatosensory cortex was preserved in rats receiving sciatic nerve ligation either with or without SCS.

To assess the temporal kinetics of cortical excitation induced by hind paw stimulation, we set regions of interest (ROIs) in the initial responses. To evaluate evoked cortical excitation in intensity, duration, and both of intensity and duration, peak amplitude, duration above 7SD, and the sum of amplitude exceeding 7SD were compared, respectively. Although a trend of reductions of this temporal kinetics, except the peak amplitude in electrical stimulation at 5 V, was observed in the application after SCS (Figure 6h), the differences were not significant.

Next, we assessed areas activated by stimulation of the hind paw. Frames obtained at 95% peak amplitude were superimposed in Figure 6i, showing that the activated areas were suppressed by SCS. Comparison of the overlapping regions in 50% of the rats between the control and SCS groups revealed that SCS induced suppression in the activated area in a concentric manner (Figure 6i). The area of response to electrical stimulation (5 V) of the hind paw of animals that underwent SCS (*n* = 6; 4.6 ± 0.8 mm^2^) was significantly lower than that of animals that did not undergo SCS (*n* = 6; 8.1 ± 1.3 mm^2^; *p* = 0.044, Student’s *t*-test; Figure 6j).

## 3. Discussion

SNI elicited mechanical hypersensitivity in the ipsilateral hind paw for at least 3 days, and the mechanical hypersensitivity was depressed by 6 h of successive SCS at 30 min and 12 h post-SCS. Moreover, successive SCS must play an important role in SNI-induced mechanical hyperalgesia, because temporary SCS does not affect the mechanical hypersensitivity in the ipsilateral hind paw. At 12 h post-SCS, SNI-induced microglial activation in laminae II of the L4 dorsal horn were significantly depressed by 6 h successive SCS. Furthermore, intraperitoneal administration of minocycline, which is a pharmacological inhibitor of microglial activation, suppressed the SNI-induced microglial activation in the laminae II and the SNI-induced mechanical hypersensitivity on day 3 post-SCS was considered as an early development phase of chronic pain. Additionally, WDR neuronal excitability in laminae II caused by innoxious and noxious mechanical stimulation of the hind paw was also enhanced on day 3 after SNI, and the WDR neuronal hyperexcitability was recovered by successive intraperitoneal administration of minocycline. Some earlier studies propose that 14 days or more post-SNI is considered chronic neuropathic pain and much more closely mimics the clinical situation because it is conceivable that the central sensitization process becomes robust from around 2 weeks onwards [16,19,20]. However, the effects of SCS on the mechanical hypersensitivity associated with microglial activation in laminae II on day 3 post-SCS and its mechanism in this study might differ considerably compared with previous reports. Together, SCS potentially suppress SNI-induced mechanical hypersensitivity in the hind paw for a period of time via suppression of the WDR neuronal hyperexcitability produced by the inhibition of SNI-induced microglial activation in laminae II in the early development phase of chronic pain, although much further work is needed.

It is well known that the lamina II of the spinal dorsal horn plays a critical role in nociceptive transmission. The activated lamina II microglia that releases some cytokines causes the central sensitization of lamina II neurons and enhances spinal nociceptive transmission [21,22,23]. Therefore, we focused the changes in the microglial activation in laminae II by SCS. Recent studies demonstrate that microglial activation, which is characterized by increased expression of Iba1 and characteristic morphologic changes, occupies an important place in the enhancement of WDR neuronal excitability via various signaling molecules following peripheral nerve injury [24], resulting in pain hypersensitivity in the hind paw. For instance, peripheral nerve injury leads to the production and release of tumor necrosis factor (TNFα) in microglia in SDH [25,26]. TNFα signaling in the transient receptor potential ion channel vanilloid 1-positive C-fiber central terminals of the sciatic nerve increases glutamate release, resulting in the enhancement of excitatory synaptic transmission in excitatory SDH interneurons in lamina II [27]. TNFα signaling also upregulates *N*-methyl-d-aspartic acid (NMDA) currents in lamina II excitatory spinal dorsal horn interneurons via the activation of the extracellular signal-regulated kinase and inhibits spontaneous firing in SDH gamma-aminobutyric acidic neurons, leading to spinal dorsal horn interneuronal hyperexcitability [28,29]. Additionally, endogenous interleukin-1β (IL-1β) released from activated microglia also upregulates presynaptic NMDA receptor function in the neuropathic pain state [30]. Furthermore, the depression of inhibitory synaptic transmissions such as GABA and glycine-induced currents by IL-1β increases spinal neuronal excitability in lamina II [31]. Thus, the signaling of IL-1β released from activated microglia in the SDH might play a key role in central sensitization by enhancement of glutamate release from the central terminals of dorsal root ganglion neurons or postsynaptic NMDA currents by the phosphorylation of the NMDA receptor as well [31,32,33]. On the basis of these reports, we presume that the depression of SNI-induced mechanical hypersensitivity attributable to SCS occurs from the inhibition of microglial activation, which enhances the production of some signaling molecules after WDR neuronal hyperexcitability.

Thus far, nothing definite is known of the inhibitory mechanism of microglial activation by SCS. Microglia in the medullary dorsal horn are activated in orofacial pathological changes, increasing cytokine release from activated microglia before the enhancement of medullary neuronal excitability in WDR neurons [34]. Carrageenan-induced activation of microglia causes the production of proinflammatory cytokines in the L4-5 spinal cord [35]. Peripheral inflammation triggers cytokine production and release from activated microglia in the spinal cord and leads to inflammatory pain hypersensitivity [36]. On the basis of these previous reports, we suggest that SCS reduces the development of neuropathic pain associated with WDR neuronal hyperexcitability via any signaling molecules from activated microglia. On the other hand, there will be little involvement of astroglial activation in SCS-induced analgesia, as there was no significant difference in the astroglial activation after SNI with SCS or post-sham stimulation.

On the other hand, recent microarray and RNA-sequencing studies separately demonstrated an upregulation of immune response and activation marker of glial cells in the spinal cord by conventional SCS [37,38]. In this study, we defined the increased density of Iba1-IR cells with large soma and short processes as microglial activation, and its activation was significantly suppressed by SCS, although the above studies define upregulation of immune-related genes or cell membrane receptors transcription as microglial activation. The difference in definition of microglial activation might be the potential reasons for the discrepancy.

In the great majority of studies, the withdrawal threshold to mechanical stimulation of hind paw was used to assess the change in mechanical pain sensitivity under pathologic conditions such as cancer, nerve injury, or inflammation [39,40,41]. However, the reduction in the withdrawal threshold might be nothing more than an exaggerated escape reflex. Therefore, we observed cortical responses to stimulation of the hind paw by using in vivo optical imaging with a voltage-sensitive dye and assessed the spatiotemporal kinetics to investigate the effects of SCS. Although temporal kinetics indicated a trend of suppression of cortical response in the application of SCS, which might reflect the reduction of inputs to the somatosensory cortex, the differences were not significant. On the other hand, we found that the area activated by electrical stimulation of the hind paw was significantly decreased by SCS. The GABAergic system in the cerebral cortex plays a critical role in the regulation of the area of excitatory propagation [42]. The SCS might induce neuronal plastic changes in the GABAergic system of the somatosensory cortex. In a future study, additional experiments, such as the whole-cell patch clamp method in slice preparation, should be performed to address this hypothesis. Although the optical imaging with a voltage-sensitive dye is appropriate to assess the degree of neural excitation with a wide field of view, the signals made with excitatory and inhibitory neural activities are observed as a summation and cannot be distinguished [43]. In a future study, furthermore experiments, such as whole-cell patch-clamp method in slice preparation, should be performed to address mechanisms underlying the suppression in area of excitatory propagation.

Several studies suggest that SCS induces an analgesic effect on pain hypersensitivity in neuropathic pain models [7,9,38,44], and a major clinical strength of SCS is that it can often dramatically reduce the intensity of neuropathic pain that other therapies have no effect on [45]. However, multiple studies have reported that the incidence of SCS complications is 30% to 40% [45,46]. The common hardware-related complications are lead failure and migration, then the common biological complications such as infection and pain over the implant [47,48]. Although pharmacological therapy does not require surgery such as the implantation of electrodes, continuous and close follow-up is always necessary to prevent side effects [49]. Clinically, appropriate treatment must be chosen for patients who suffer from the neuropathic pain.

Fourteen days post-lesion is considered chronic neuropathic pain in most pre-clinical studies where chronic neuropathic pain has been defined as that which persisted 14 days post-lesion; 30–40 min SCS stimulation paradigms have been tested in these studies [50,51,52]. On day 3 after peripheral nerve injury, the central sensitization process is not as robust as it is after 14 days. Furthermore, the effects and mechanism of SCS in this study might differ considerably. Therefore, due to differences in study design, it is impossible to directly compare our findings with those of clinical studies and further research is recommended.

## 4. Materials and Methods

### 4.1. Animals

This study was performed by using male Sprague-Dawley rats (6-weeks-old, *n* = 81; Japan SLC, Shizuoka, Japan) that were bred well under appropriate conditions (room temperature: 23 °C; light-dark cycle: every 12 h; ad libitum access to water and food). The Animal Experimentation Committee of Nihon University approved all procedures in the study (AP16D003-2 (9/23/2016), AP18DEN007-1 (4/27/2018)). The experiments were conducted by the guidelines issued by the International Association for the Study of Pain [53]. In all procedures, animal suffering and the number of animals were maximally reduced. Appendix A shows the schematics of the timeline of the experimental interventions.

### 4.2. Spared Nerve Injury Model Preparation and SCS Electrode Implantation

Rats were anesthetized with intraperitoneal (i.p.) butorphanol (2.5 mg/kg, Meiji Seika Pharma, Tokyo, Japan), medetomidine (0.375 mg/kg, Zenoaq, Koriyama, Japan), and midazolam (2.0 mg/kg, Sandoz, Tokyo, Japan). The spared nerve injury (SNI) model was generated as previously described by Decosterd and Woolf [5]. Briefly, after pelvifemoral skin incision and muscle detachment, the common peroneal and tibial nerves were carefully isolated and dissected by scissors without any stress of the intact sural nerve.

In brief, back-skin incision and muscle detachment were performed, and the spinal column at the L1-4 levels was exposed by ablation of muscles attached to vertebrae with forceps under the deep anesthesia. After the SNI, the lead for SCS (TU216-007; Unique Medical, Tokyo, Japan) (Appendix A) was paraspinally placed at the L4-S1 spinal cord without damaging the vertebrae or dura at all, with a cathode being installed in L4, and an anode being installed in S1. Therefore, the lead can be stimulating the L4-S1 spinal cord. The incised back-skin was closely sutured such that a part of the proximal end of the electrode was embedding subcutaneously. The incised skin and detached muscle were sutured using 4–0 silk with the treatment of a local anesthetic ointment (2% xylocaine jelly; Aspen, Tokyo, Japan), and buprenorphine (0.9 μg, Otsuka, Tokyo, Japan) was administered subcutaneously for pain management.

### 4.3. Plantar Mechanical Sensitivity

Before (naive) and on day 3 after electrode implantation, rats were placed on a mesh-floor plastic box and allowed to adapt to their environments for 30 min before the testing. The plantar mechanical sensitivity was assessed using von Frey filaments pressed against the plantar surface in ascending order (pressures: 1.0, 1.4, 2.0, 4.0, 6.0, 8.0, 10, 15, 26, 30, 40, 50, and 60 g). The paw withdrawal threshold (PWT) was determined as the lowest intensity evoking withdrawal response more than three times out of five stimuli (duration: 5 s). For each trial, the filament was applied at 1 s intervals. A cutoff (60 g) was defined to avoid tissue injury. The interval between each mechanical stimulus was set at >3 min to avoid sensitization of the receptive field by frequent stimulus. The timing was based on previous studies [54,55,56]. Measurement of the PWT was performed under the same conditions and conducted in a randomized, experimenter-blinded design.

### 4.4. SCS

One hour after measurement of the PWT, an external neurostimulator (# 37022, Medtronic, Dublin, Ireland) that can produce electrical pulses following the SCS protocol (#8840, Medtronic) was connected with the proximal end of the electrode after cutting the 4–0 silk used to suture the incised skin, and pulling out the proximal end of electrode embedded subcutaneously under inhalation anesthesia with 2% isoflurane (Mylan, Southpointe, PA, USA). First, the motor threshold (MoT) was determined as the minimum amplitude of electrical stimulation (frequency: 4 Hz, duration: 240 µs) that evokes mid-lower trunk or hind limb muscle contraction under inhalation anesthesia (2% isoflurane). After determination of MoT, SCS (frequency: 60 Hz, duration: 240 µs, 80% MoT) or sham stimulation (0 mA) was performed for 6 h. The PWTs were measured at 30 min and 12 h after the completion of SCS as described above in a randomized, experimenter-blinded design. The timing of the recordings (30 min and 12 h post SCS) and SCS duration (6 h) were based on the pervious study [57].

### 4.5. Effect of Minocycline Administration on Plantar Mechanical Sensitivity Following SNI

A daily intraperitoneal minocycline hydrochloride (30 mg/kg/day, Merck, Darmstadt, Germany) or vehicle (saline) administration was performed for 4 days (day 0, 1, 2, and 3) after SNI. The dosage and administration method of minocycline were determined on the basis of previous studies [44,58,59]. On day 3 following SNI, the PWT was measured according to the behavioral testing method described above in a randomized (minocycline or vehicle), experimenter-blinded design.

### 4.6. Immunohistochemistry

At 12 h after the completion of SCS or sham stimulation, rats were perfused with 4% paraformaldehyde dissolved in 0.1 M phosphate buffer followed by physiological saline under deep anesthesia using sodium pentobarbital (i.p., 50 mg/kg). Sham-injured rat-treated sham stimulation were also perfused in a similar manner. The L4-S1 spinal cord was dissected and immersed in the above fixative for 24 h. Then, spinal horizontal sections were cut with a freezing microtome (thickness: 30 µm) following immersion in 20% sucrose for 12 h, and the three free-floating sections corresponding to L4 were collected in 0.01 M phosphate-buffered saline (PBS). Because activated microglia showed an intense increase in Iba1 expression in various pathological states, we introduced Iba1 antibody for immunohistochemical analysis to evaluate whether microglial cells were activated [60]. Moreover, we introduced glial fibrillary acidic protein (GFAP) antibody for immunohistochemical analysis to evaluate whether astroglial cells were activated. The sections were incubated in goat anti-Iba1 polyclonal antiserum corresponding to the C-terminus of Iba1 (1:500, 019-19741; Wako, Osaka, Japan) or mouse anti-GFAP monoclonal antibody (1:500, MAB360; Merck, Darmstadt, Germany) in 0.3% TritonX-100 in 0.01M PBS with 4% normal donkey serum overnight at 4 °C. After rinsing with 0.01M PBS, the sections were incubated in Alexa Fluor 488 donkey anti-rabbit IgG diluted in 0.01M PBS (1:200, Merck) or Alexa Fluor 568 donkey anti-mouse IgG diluted in 0.01M PBS (1:200, Merck) for 2 h at 23 °C. After rinsing with 0.01M PBS, the sections were coverslipped in mounting medium (Thermo Fisher Scientific, Waltham, MA, USA), and Iba1-immunoreactive (IR) cells were examined by using fluorescence microscopy (Keyence BZ9000; Keyence, Osaka, Japan). The immunoreactivities of Iba-1 and GFAP showing an intensity twofold greater than the average background were considered positive for Iba-1 and GFAP immunoreactivity. The densities of Iba1-IR and GFAP-IR cells were measured in laminae II (313.37 × 313.37 µm^2^) of L4 SDH by using a computer-assisted imaging analysis system (ImageJ 1.37; National Institutes of Health, Bethesda, MD, USA). The densities of Iba1-IR and GFAP-IR cells were defined in the areas in laminae II (313.37 × 313.37 µm^2^) of L4 SDH occupied by the Iba1-immunoproducts in laminae II (313.37 × 313.37 µm^2^) of L4 SDH. All immunohistochemical examinations were conducted under randomized conditions (ipsilateral or contralateral, SCS or sham stimulation) and the maximum degree of blinding.

### 4.7. Western Blotting

On day 3 after SNI or sham treatment, the rats were perfused with physiological saline under the i.p. deep anesthesia with the solution described above. Immediately, the L4 SDH ipsilateral to SNI or sham treatment was removed and homogenized in ice-cold lysis buffer (137 mM NaCl; 20 mM Tris-HCl, pH 8.0; 1% NP40; 10% glycerol; 1 mM phenylmethylsulfonyl fluoride; 10 μg/mL aprotinin; 1 g/mL leupeptin; 0.05 mM sodium vanadate). The homogenate was centrifuged, and the supernatants were extracted. The protein concentration of the supernatants was determined using a protein assay kit (Bio-Rad, Hercules, CA, USA). The supernatants were heat-denatured in Laemmli sample buffer solution (Bio-Rad), and the samples with protein adjusted to 30 μg were subjected to electrophoresis on 10% sodium dodecyl sulfate–polyacrylamide gel electrophoresis for protein separation. The samples were transferred to a polyvinylidene difluoride membrane (Trans-Blot Turbo Transfer Pack; Bio-Rad) utilizing Trans-Blot Turbo (Bio-Rad). The membrane was rinsed with Tris-buffered saline mixed with 0.1% Tween 20 (TBST, Bio-Rad) and incubated in 3% bovine serum albumin (Bovogen, Essendon, Australia). The membrane was then incubated with rabbit anti-Iba1 polyclonal antibody (1:1,000; Wako, Tokyo, Japan) diluted in TBST with 3% BSA overnight at 4 °C. Then, the horseradish peroxidase-conjugated rabbit anti-rabbit antibody (Jackson Immuno Research, West Grove, PA, USA) was incubated for 2 h at room temperature. Protein binding antibody was detected using Western Lightning ELC Pro (PerkinElmer, Waltham, MA, USA) and visualized using a ChemiDoc MP system (Bio-Rad). Using β-actin antibody (1:200; Santa Cruz, Santa Cruz, CA, USA) following removing bound protein by a stripping reagent (Thermo Fisher Scientific, Waltham, MA USA), the protein level was normalized to β-actin.

### 4.8. Single Neuronal Recording

At 12 h after completion of SCS or sham stimulation, the rats were strictly secured in the stereotaxic frame under the aforementioned deep anesthesia. The L4-5 SDH was exposed following a small laminectomy. Intravenous injection of pancuronium bromide (0.6 mg/kg/h, Merck) was performed to produce muscular relaxation, and the rats were artificially ventilated. During the recording, end-tidal CO_2_ concentration (3.5% to 4.5%), body temperature (37 °C), and heart rate remained steady under artificial respiration with oxygen (2 L/min) and isoflurane (2.0%). In this setting, enamel-coated tungsten microelectrodes (FHC, Bowdoin, ME, USA) were advanced into the lamina II in the L4-5 SDH, and single neuronal activities were recorded. The graded mechanical stimuli were applied to the receptive field of the hind paw identified by gentle brush stimulus. The neuron that responded to both nonnoxious and noxious mechanical stimuli and increased its firing frequency as stimulus intensity increased was defined as wide dynamic range (WDR) neuron. The neuron that responded exclusively to noxious mechanical stimulation was defined as a nociceptive specific neuron [61]. WDR neurons responding to innocuous stimulation and, to a greater degree, noxious stimulation of the hind paw were examined in this study. After the identification of a WDR neuron, mechanical stimuli were applied to the hind paw following the recording of the neuron’s background activity for 30 s. For low-intensity mechanical stimulation of the neuronal mechanoreceptive field, graded stimuli with von Frey filaments (1.0, 6.0, 15, 26, 60 g) and brushing with a nylon hair brush were applied for 5 s at 10 s intervals. High-intensity (pinch) stimulation with pinch produced by a small arterial clip was also applied for 1 min. The mechanical stimuli were applied 5 times for 5 s at a 1 min interval. This single neuronal activity was amplified using a differential amplifier (Nihon Koden, Tokyo, Japan) and stored in the microcomputer hard disk. Spikes were sorted, and spike frequencies were analyzed using the Spike II software (CED, Cambridge, UK). Neuronal responses were defined when the mean firing frequency was >2 standard deviations (SD) of the background activity, and the mean number of spikes elicited by mechanical stimuli applied for 5 s was examined at each stimulus intensity.

### 4.9. In Vivo Optical Imaging With a Voltage-Sensitive Dye

In optical imaging experiments, 12 rats were divided into two groups, that is, control group (rats received SNI but not SCS; *n* = 6) and SCS group (rats received SNI and SCS; *n* = 6). Optical imaging using a voltage-sensitive dye (RH-1691, Optical Imaging, New York, NY, USA) was performed as previously described [42,62,63,64]. Briefly, the rats received an atropine methyl bromide injection (5 mg/kg, i.p.), were anesthetized with urethane (1.5 g/kg, i.p., Sigma-Aldrich, St. Louis, MO, USA), and were maintained at approximately at 37 °C by using a rectal probe and a heating pad (BWT-100, Bio Research Center, Osaka, Japan). Additional urethane was administered depending on the toe pinch reflex, and the observation of cortical responses was performed under spontaneous breathing. Rats were fixed to a stereotaxic snout frame, and a craniotomy was performed to expose the right somatosensory cortex. We carefully resected the dura matter. After dura resection, RH-1691 (1 mg/mL) in 0.9% saline was applied to the cortical surface for 1 h. Then, the cortical surface was rinsed with saline and covered with 1% agarose (Agarose Low EEO, Sigma-Aldrich) dissolved in Ringer’s solution and a glass coverslip. Bipolar enamel-coated copper wire electrodes (diameter = 80 μm) were inserted into the ulnar side of the palm of the left forepaw and hind paw. To assess the cortical responses to the forepaw and hind paw stimulation, 5 voltage pulses (100 μs duration, 2.5 and 5 V) at 50 Hz were applied using a stimulator unit (STG2008, Multi Channel Systems, Reutlingen, Germany). As was the case with the above-mentioned anesthesia with urethane, the electrical stimulation-induced spinal reflex was not observed visually. The cortical surface was illuminated through a 632 nm excitation filter with a dichroic mirror using a tungsten halogen lamp (CLS150XD, Leica Microsystems, Wetzlar, Germany). Fluorescent emission was captured through an absorption filter (λ > 650-nm longpass, Andover, Salem, MA, USA). Changes in RH-1691 fluorescence intensity in response to the electrical stimulation of forepaw and hind paw were measured at 250 Hz for 1000 ms, including a baseline period of 50 ms, by using a CCD camera system (MiCAM02, Brainvision, Tokyo, Japan) mounted on a stereomicroscope (Leica Microsystems). The CCD camera had a 6.4 × 4.8 mm^2^ imaging area (184 × 124 pixels). To correct signals induced by acute bleaching of the dye, the fluorescence intensity without stimulation was subtracted from each recording. Electrical stimulation was performed with an interstimulus interval of 20 s, and 40 images were averaged to improve the signal-to-noise ratio.

A software program (Brain Vision Analyzer; Brainvision) was used to process and analyze optical signals. Changes in the intensity of fluorescence (*ΔF*) of each pixel relative to the initial intensity of fluorescence (*F*) were calculated (*ΔF/F*), and the ratio was processed with a spatial filter (9 × 9 pixels). A significant response was defined as a signal exceeding seven times the SD of the baseline period. A region of interest (ROI), a circle consisting of 77 pixels (≈0.1 mm^2^), was set on the basis of the initial response. Peak amplitude was defined as the maximum amplitude of optical response in the ROI. Durations above 7 SD were defined as the duration of a significant response. To evaluate activated areas, we used the frames in which the optical signal first exceeded 95% of the maximum amplitude in the ROI [63].

### 4.10. Statistical Analysis

For comparisons of plantar mechanical sensitivity, Iba1 Western blotting and immunoreactivity, and lamina II neuronal activity, Mann–Whitney *U* tests, one-way or two-way repeated-measures analysis of variance (ANOVA) followed by Tukey’s or Bonferroni’s multiple-comparison tests, and Student’s *t*-tests were performed for statistical analyses by using Prism 7 (GraphPad Software, San Diego, CA, USA), appropriately. In a comparison of spatiotemporal kinetics, Student’s *t*-test was used, following appropriate normality and equal variance tests. SigmaStat software (ver. 4.0, Systat Software, San Jose, CA, USA) was used for statistical analyses. In the comparison of plantar mechanical sensitivity, data were indicated as medians and 25th to 75th percentiles, with the minimum and maximum values. Data for Iba1 immunoreactivity, lamina II neuronal activity, and spatiotemporal kinetics were expressed as the means ± standard error (SEM). A *p*-value of less than 0.05 was considered statistically significant.

## 5. Conclusions

Overall, our data suggest that SCS could persistently suppress the microglial activation in laminae II following SNI, resulting in the depression of SNI-induced mechanical hypersensitivity in the hind paw relevant to the inhibition of SNI-induced WDR neuronal hyperexcitability. Moreover, electrical stimulation of hind paw-induced excitatory propagation in the somatosensory cortex was significantly decreased by SCS, suggesting that SCS decreased the pain hypersensitivity at the CNS level. Further studies are needed to elucidate the analgesic mechanisms of SCS in humans.

## Figures and Tables

**Figure 1 ijms-21-02390-f001:**
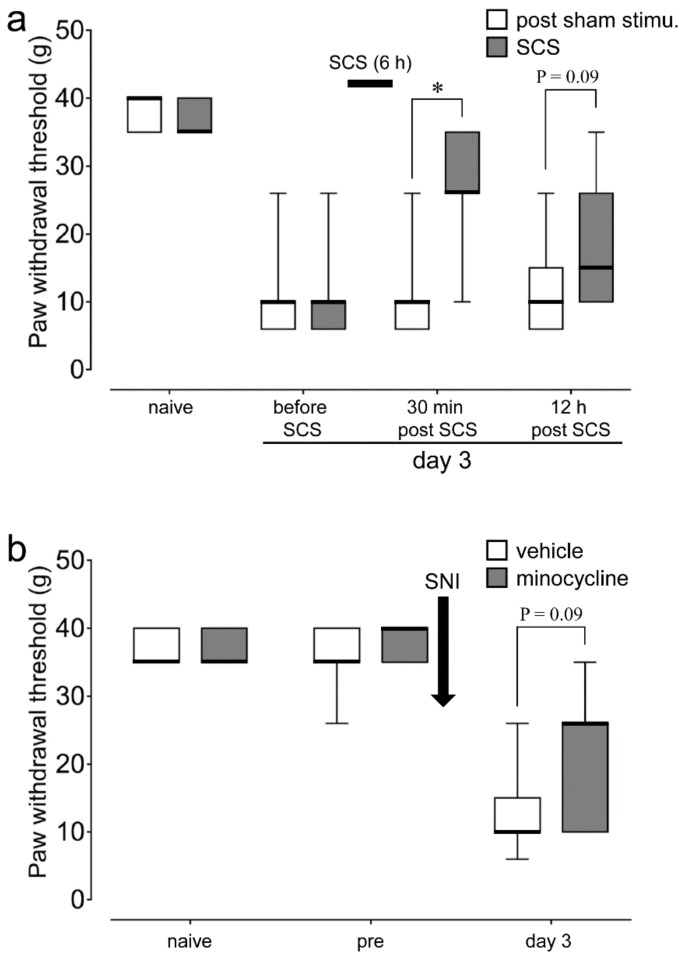
Effect of spinal cord stimulation (SCS) or minocycline on plantar mechanical hypersensitivity following spared nerve injury (SNI). (**a**) Paw withdrawal threshold (PWT) before SCS and at 30 min and 12 h after 6 h of successive SCS or sham stimulation on day 3 following SNI and PWT in naive rats. post-sham stimu.: post-sham stimulation. * *p* < 0.05 vs. post-sham stimu. (Mann–Whitney *U* test). (**b**) PWT before and on day 3 after SNI with once-daily intraperitoneal (i.p.) vehicle (saline) or minocycline (30 mg/kg/day) administration and PWT in naive rats (Mann–Whitney *U* test).

**Figure 2 ijms-21-02390-f002:**
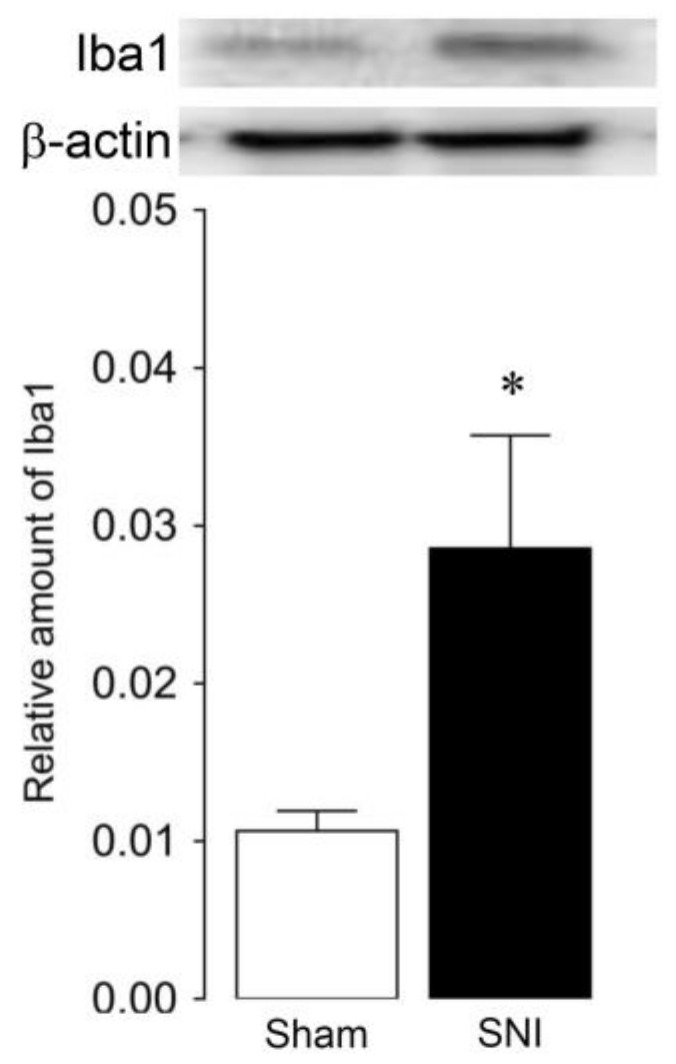
The relative amount of Iba1 protein in the ipsilateral L4 dorsal horn on day 3 following SNI. Error bars indicate standard error of the mean (SEM). **p* < 0.05 sham. (Student’s *t*-test).

**Figure 3 ijms-21-02390-f003:**
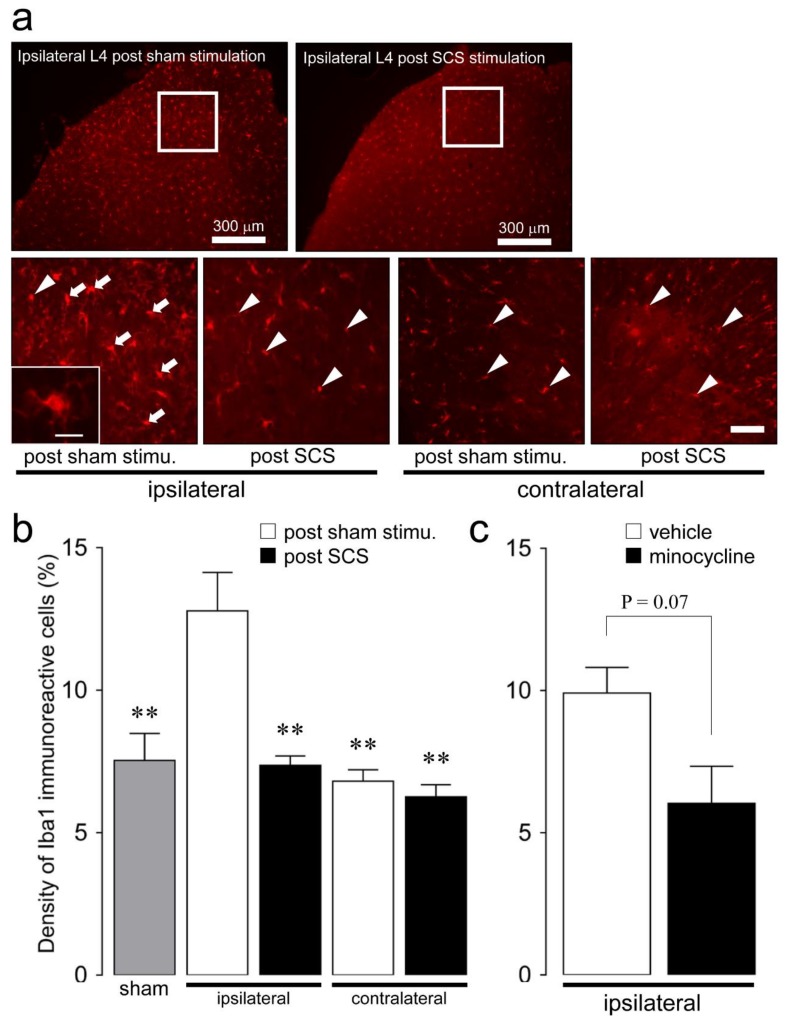
Changes in the L4 dorsal horn microglial activation on day 3 following SNI depending on SCS. (**a**) Iba1-immunoreactive (IR) cells in the bilateral L4 dorsal horn on day 3 following SNI with SCS or sham stimulation. White frame indicates the density analysis area. Arrows indicate activated Iba1-IR cells. Arrowhead indicates non-activated Iba1-IR cells. Scale bar: 100 µm. Inlet in (**a**) low and high magnification fluorescence micrograph of the activated Iba1-IR cells. Scale bar: 50 µm. (**b**) The bilateral density of Iba1-IR cells in the L4 dorsal horn on day 3 after SNI with SCS or sham stimulation, and in sham-injured rat-treated sham stimulation. Error bars indicate SEM. ** *p* < 0.01 vs. post-sham stimulation (one-way ANOVA with Tukey’s multiple-comparison test). (**c**) The ipsilateral density of Iba1-IR cells in the L4 dorsal horn on day 3 after SNI with once-daily i.p. vehicle (saline) or minocycline (30 mg/kg/day) administration. Error bars indicate SEM. (Student’s *t*-test).

**Figure 4 ijms-21-02390-f004:**
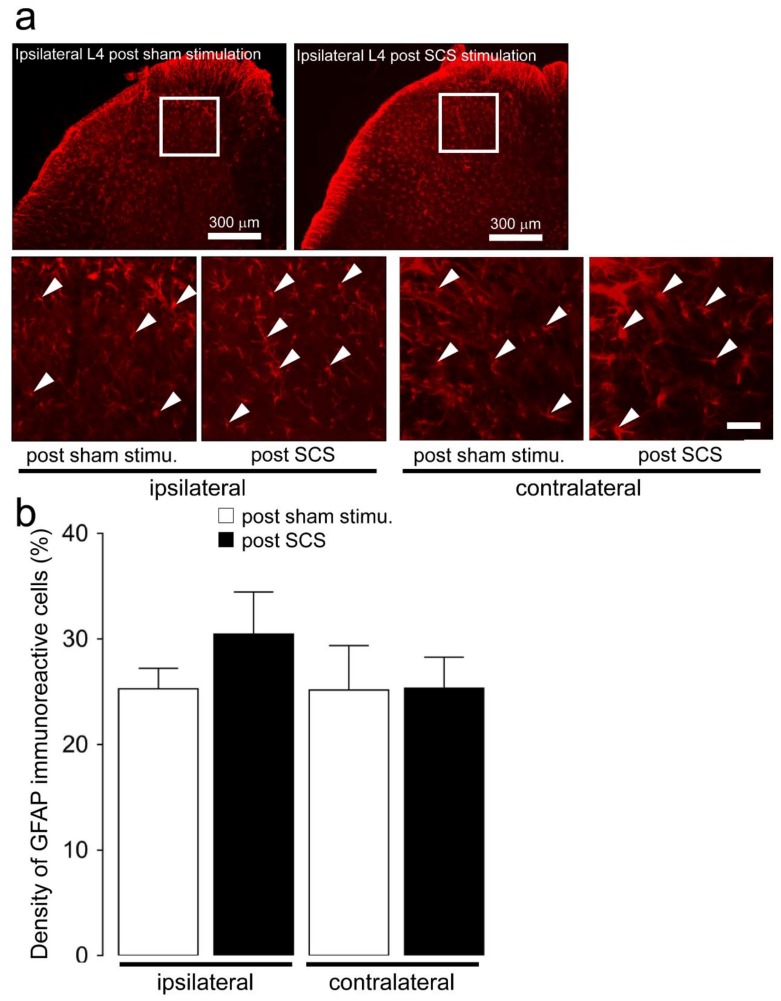
Changes in the L4 dorsal horn astroglial activation on day 3 following SNI depending on SCS. (**a**) Glial fibrillary acidic protein (GFAP)-IR cells in the bilateral L4 dorsal horn on day 3 following SNI with SCS or sham stimulation. Arrows indicate activated GFAP-IR cells. Scale bar: 100 µm. (**b**). The bilateral density of GFAP-IR cells in the L4 dorsal horn on day 3 after SNI with SCS or sham stimulation. Error bars indicate SEM.

**Figure 5 ijms-21-02390-f005:**
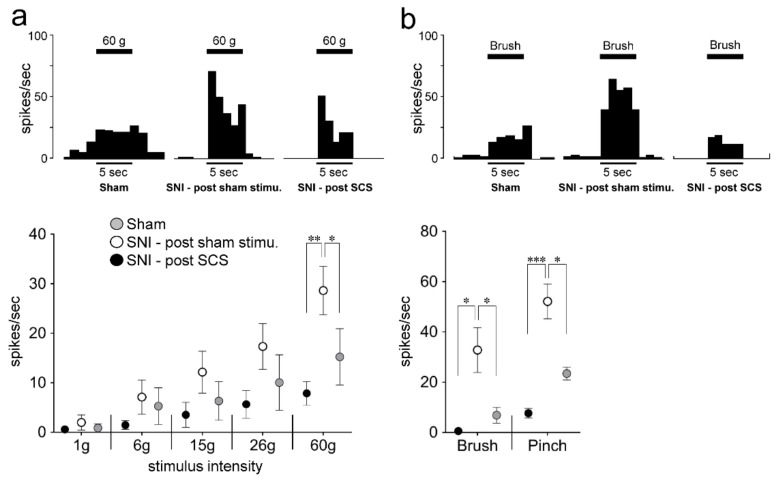
The effect of SCS on mechanical-evoked responses of wide dynamic range (WDR) neurons in lamina II of the dorsal horn in L4-S1 on day 3 following SNI with SCS or sham stimulation. The raw traces of WDR neurons responses and the frequency of WDR neuronal spikes in response to mechanical stimuli by von Frey filament (**a**) and brush or pinch stimuli (**b**) on day 3 after SNI with SCS or sham stimulation. Error bars indicate SEM. **p* < 0.05, ** *p* < 0.01, *** *p* < 0.001 vs. post-sham stimulation (two-way ANOVA with repeated measures followed by Bonferroni’s multiple-comparison tests or Student’s *t*-test).

**Figure 6 ijms-21-02390-f006:**
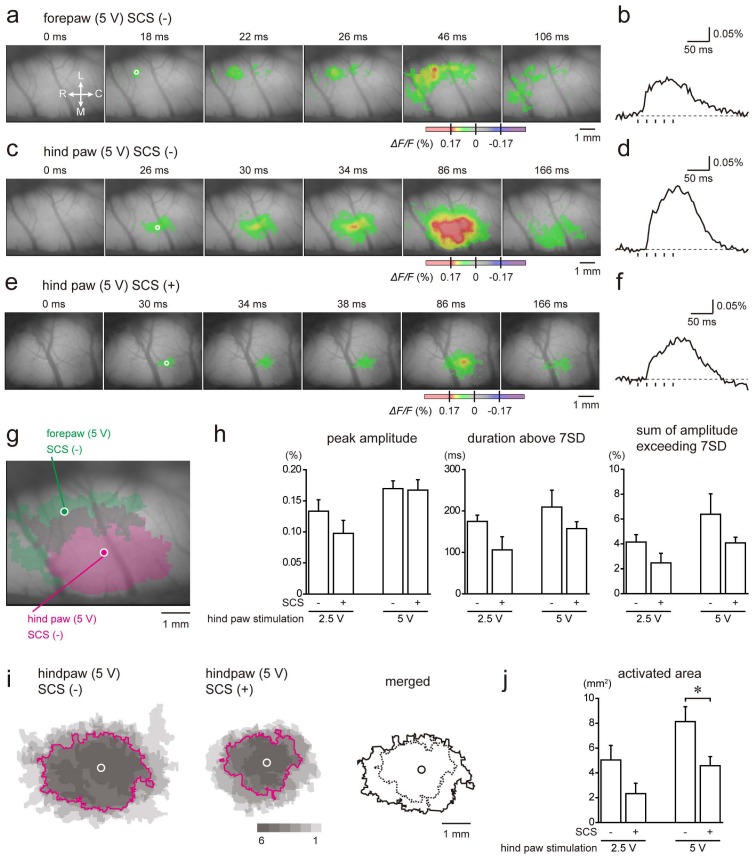
Responses in the right somatosensory cortex to electrical stimulation applied to the left forepaw and hindpaw in the absence (-) or presence (+) of SCS in the SNI model. An example of excitatory propagation (**a**) and the trace of optical signals (**b**) during stimulation of forepaw in a rat without SCS. In (**a**), the amplitude of the optical signal (*ΔF*/*F*) is color-coded, and the time from the onset of the stimulation of the hindpaw is shown at the top of each panel. The trace in (**b**) is obtained in the region of interest (ROI) indicated by the white circle in (**a**). Vertical lines in (**b**) indicate the applied 5 voltage stimuli. C, caudal; L, lateral; M, medial; R, rostral. An example of excitatory propagation (**c**) and the trace of optical signals (**d**) during stimulation of the hind paw in a rat without SCS is shown. An example of excitatory propagation (**e**) and the trace of optical signals (**f**) during stimulation of the hind paw in a rat with SCS is shown. (g) The initial responses (circles) and activated areas (colored areas) in forepaw stimulation (green) and hind paw stimulation (magenta) without SCS are shown. Note that the region responding to hind paw stimulation was medial and caudal to the region responding to forepaw stimulation. The data shown in (**a**–**d**), and (**g**) were obtained from the same animal. (**h**). Summary of peak amplitude, duration above 7SD, and the sum of amplitude exceeding 7SD obtained in the initial responses. Note that there are no significant differences between the control group (*n* = 6) and SCS group (*n* = 6). (**i**). Initial responses (circle) are aligned and activated areas in the control and SCS groups are superimposed. The number of overlapping responses is represented by the density of color. Magenta outlines indicate the overlapping areas in four of six rats. In the right panel, the outlines were merged. Solid and dashed lines indicate control and SCS groups, respectively. (**j**). Comparison of activated areas in response to hind paw stimulation without and with SCS. **p* < 0.05. (Student’s *t*-test).

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
