# Peer review of "Suppression of Superficial Microglial Activation by Spinal Cord Stimulation Attenuates Neuropathic Pain Following Sciatic Nerve Injury in Rats"

_ijms, 2020, doi:10.3390/ijms21072390_

Round 1
Reviewer 1 Report
Approve for publication
Author Response
Thank you very much for reviewing our paper.
Reviewer 2 Report
This research has been conducted to identify mechanisms underlying analgesic effects of SCS in a rat model of nerve injury. Findings presented in the manuscript have provided evidence that plantar mechanical hypersensitivity and enhanced microglial activation were suppressed by SCS. In addition, L4 dorsal horn nociceptive neuronal hyperexcitability was suppressed by SCS. In vivo optical imaging has presented that electrical stimulation of the hind paw-activated areas in the somatosensory cortex was also suppressed by SCS. Authors – based on these findings - proposed that SCS could suppress plantar SNI-induced neuropathic pain via inhibition of microglial activation in the L4 dorsal horn, which is involved in spinal neuronal hyperexcitability.
There are a couple of points that authors are encouraged to clarify further:
In the introduction and subsequently in the discussion, add the timing of hyperactive states of microglia based on the literature and the fact that minocycline needs to be administered daily for almost 14 days to present an effect. It brings then a question on how ip injection of this agent has been performed for 3 days and it has been sufficient for suppression of microglial activity? Is this the protocol for minocycline administration? If not, please make it clear for timing and duration. Choice of dose for administration is also important to add, whether it is based on own pilot studies or based on literature recommendation.
Is there any adverse effect or complications with the use of SCS or minocycline? Authors are encouraged to point out to both strengths and limitations of use of pharmacological versus non-pharmacological strategies.
In fig legends, please add the dose of minocycline administered. For SCS, this can be added as the characteristics of SDS application.
In Fig 2, the panel is with a poor resolution for Iba1. Please provide a clear protein band image.
What is vehicle for minocycline? Add in the method and in fig legends, for example fig 3.
In method, add the choice of timing for the records, 6h, 3 days, etc. Either these are based on literature that needs a reference, or based on own pilot tests from the authors of this study that must be added.
In page 11, TNFα must be corrected. It is with a strange symbol presented instead of α, if that is the purpose of authors.
Choice of only including “male” animals, needs justification.
Choice of method for von Frey is not clear. Please add whether it has followed the standard method of ascending or descending or another method.
Blinding and randomization in this study must be clearly explained in the method section. What about randomization of the ipsi lateral versus contralateral, was that randomized and balanced or followed a fixed protocol, for left and right?
In the statistics, add P value set for significant level.
Include limitations of the present study.
Author Response
This research has been conducted to identify mechanisms underlying analgesic effects of SCS in a rat model of nerve injury. Findings presented in the manuscript have provided evidence that plantar mechanical hypersensitivity and enhanced microglial activation were suppressed by SCS. In addition, L4 dorsal horn nociceptive neuronal hyperexcitability was suppressed by SCS. In vivo optical imaging has presented that electrical stimulation of the hind paw-activated areas in the somatosensory cortex was also suppressed by SCS. Authors – based on these findings - proposed that SCS could suppress plantar SNI-induced neuropathic pain via inhibition of microglial activation in the L4 dorsal horn, which is involved in spinal neuronal hyperexcitability.
There are a couple of points that authors are encouraged to clarify further:
In the introduction and subsequently in the discussion, add the timing of hyperactive states of microglia based on the literature and the fact that minocycline needs to be administered daily for almost 14 days to present an effect.
# Minocycline does not need to be administered daily for almost 14 days to present an effect, a single administration of minocycline rapidly (within one hour) and completely reverses the mechanical hypersensitivity following peripheral nerve injury for 3 hours and the inhibitory effect fades after 5 hours (Neurosci Bull. 34(1):98-108, 2018). As regards the timing of hyperactive states of microglia, we have added the sentences as follows;
L66-69. “In SDH, the increased Iba1 immunoreactivity, which indicates the hyperactive states of microglia, was reported to be detected as early as 20 min, peaked on day 3, and remained at a significant level on day 7 following peripheral nerve injury [14].”
It brings then a question on how ip injection of this agent has been performed for 3 days and it has been sufficient for suppression of microglial activity? Is this the protocol for minocycline administration? If not, please make it clear for timing and duration.
# We have already reported that a once-daily intraperitoneal minocycline (for 4 days, 30 mg/kg/day) is sufficient for suppression of microglial activity and neuropathic pain (Brain Res.1451:74-86. 2012). To make it clear for timing and duration of minocycline administration, we have modified the sentences as follows;
L363-365 “A daily intraperitoneal minocycline hydrochloride (30 mg/kg/day, Merck, Darmstadt, Germany) or vehicle (saline) administration was performed for four days (day 0, 1, 2, and 3) after SNI.”
Choice of dose for administration is also important to add, whether it is based on own pilot studies or based on literature recommendation.
# Based on these literatures (Brain Res.1451:74-86. 2012; Mol Pain.10:47. 2014; Clin Exp Pharmacol Physiol. 42(1):94-101. 2015; the dosage and administration method of minocycline were determined. We have added the sentence as follows;
L365-366. “The dosage and administration method of minocycline were determined on the basis of previous studies [44, 58, 59].”
Is there any adverse effect or complications with the use of SCS or minocycline? Authors are encouraged to point out to both strengths and limitations of use of pharmacological versus non-pharmacological strategies.
# We have added the paragraph as follows in the Discussion section.
L295-303. “Several studies suggest that SCS induces an analgesic effect on pain hypersensitivity in neuropathic pain models [7, 9, 38, 44], and a major clinical strength of SCS is that it can often dramatically reduce the intensity of neuropathic pain which other therapies have no effect on [45]. However, multiple studies have reported that the incidence of SCS complications is 30% to 40% [45, 46]. The common hardware-related complications are lead failure and migration, then the common biological complications such as infection and pain over the implant [47, 48]. Although pharmacological therapy does not require surgery such as the implantation of electrodes, continuous and close follow-up is always necessary to prevent side effects [49]. Clinically, appropriate treatment must be chosen for patients who suffer from the neuropathic pain.”
In fig legends, please add the dose of minocycline administered. For SCS, this can be added as the characteristics of SDS application.
# In fig legends, we have added the dose of minocycline administered as follows;
L104-105. “PWT before and on day 3 after SNI with once-daily i.p. vehicle (saline) or minocycline (30 mg/kg/day) administration”
L135-137. “The ipsilateral density of Iba1-IR cells in the L4 dorsal horn on day 3 after SNI with once-daily i.p. vehicle (saline) or minocycline (30 mg/kg/day) administration. Error bars indicate SEM. (Student’s t-test).”
In Fig 2, the panel is with a poor resolution for. Please provide a clear protein band image.
# We have replaced the Iba1band image with clear Iba1 band image in Figure 2.
What is vehicle for minocycline? Add in the method and in fig legends, for example fig 3.
# We have described the component of vehicle for minocycline in the method and in fig legends.
In method, add the choice of timing for the records, 6h, 3 days, etc. Either these are based on literature that needs a reference, or based on own pilot tests from the authors of this study that must be added.
# Numerous previous studies (Int J Clin Exp Pathol. 12(8):2898-2908. 2019; Neuromolecular Med. doi: 10.1007/s12017-019-08581-3. 2019; Sci Rep. 9(1):14664. 2019; Exp Neurobiol. 28(4):516-528. 2019) have indicated that neuropathic pain is induced reliably on day 3 after sciatic nerve injury in rats. Therefore, we have set the timing for the record on day 3 following SNI, to assess the effect of SCS or minocycline on the neuropathic pain.
In this study, the choices of timing for the records (30 min and 12 h post SCS) and SCS duration (6 hours) are based on the literature (Anesth Analg. 119(1):186-95. 2014).
We have added the sentences as follows;
L347-348. “The timing was based on previous studies [54, 55, 56].”
L348-349. “The timing of the recordings (30 min and 12 h post SCS) and SCS duration (6 hours) were based on the pervious study [57].”
In page 11, TNFα must be corrected. It is with a strange symbol presented instead of α, if that is the purpose of authors.
# We have corrected.
Choice of only including “male” animals, needs justification.
# Recent studies show that male and female immune systems including microglial and astroglial behavior react differently during abnormal pain, and the efficacy of immune targeted pain treatments depends on sex (J Neurosci Res. 95:500–8. 2017; Nat Neurosci 18:1081–3. 2015). Therefore, first of all, we have focused the involvement of superficial microglial deactivation by spinal cord stimulation in neuropathic pain following sciatic nerve injury in male rats in this study. As the next step, we have planned the study using female rats.
Choice of method for von Frey is not clear. Please add whether it has followed the standard method of ascending or descending or another method.
# We have chosen the method used in our previous studies (Neurosci Res. in press. 2020; Int J Mol Sci. 20. 2019; J Neurosci. 33, 7667-80. 2013). To provide a full account of the method used in this study, we have added the sentences as follows;
L345-348. “For each trial, the filament was applied at 1 s intervals. A cutoff (60 g) was defined to avoid tissue injury. The interval between each mechanical stimulus was set at >3 min to avoid sensitization of the receptive field by frequent stimulus. The timing was based on previous studies [54, 55, 56].”
Blinding and randomization in this study must be clearly explained in the method section. What about randomization of the ipsi lateral versus contralateral, was that randomized and balanced or followed a fixed protocol, for left and right?
# To explain the blinding and randomization in this study, we have modified the sentences as follows;
L348-349. “Measurement of the PWT was performed under the same conditions and conducted in a randomized, experimenter-blinded design.”
L359-360. “The PWTs were measured at 30 min and 12 h after the completion of SCS as described above in a randomized, experimenter-blinded design.”
L366-368. “On day 3 following SNI, the PWT was measured according to the behavioral testing method described above in a randomized (minocycline or vehicle), experimenter-blinded design.”
L395-396. “All immunohistochemical examinations were conducted under randomized conditions (ipsilateral or contralateral, SCS or sham stimulation) and the maximum degree of blinding.”
In the statistics, add P value set for significant level.
# We have added the sentences as follows;
L492-493. “A p-value of less than 0.05 was considered statistically significant.”
Include limitations of the present study.
# To include limitations of the present study, we have added the sentences as follows;
L304-310. “Fourteen days post-lesion is considered chronic neuropathic pain in most pre-clinical studies where chronic neuropathic pain was defined as that which persisted 14 days post-lesion; 30-40 min SCS stimulation paradigms have been tested in these studies [50, 51, 52]. On day 3 after peripheral nerve injury, the central sensitization process is not as robust as it is after 14 days. Furthermore, the effects and mechanism of SCS in this study might differ considerably. Therefore, due to differences in study design, it is impossible to directly compare our findings with those of clinical studies and further research is recommended.”
Reviewer 3 Report
The manuscript describes an original research performed in rats focusing on the mechanisms underlying spinal cord stimulation (SCS)–induced analgesic effect on neuropathic pain. Authors investigated hypothesis that SCS-induced analgesic effect on neuropathic pain may be due to suppression of the spared nerve injury (SNI)-induced microglial activation in spinal cord. This idea is supported by results from experiments using immunohistochemistry, electrophysiological and optical imaging methods. Beneficial effect of SCS on neuropathic pain is associated with microglial inactivation which is accompanied by suppression of spinal neurons hyperexcitability and reduction of SNI-activated area within somatosensory cortex.
Remarks and comments:
1) Line 57-59 “..At present, the exact mechanisms underlying the analgesic effect attributable to SCS remain poorly understood…”
Has SCS only analgesic effect? Or SCS exerts dual action on pain in humans and animals? Add this information in Introduction or Discussion.
2) Line 89-90 “..On the other hand, SCS for thirty minutes did not recover the decrement of PWT at 3 and 6 hours post-SCS (post-sham stimulation (n = 7), SCS (n = 7)) …”
Line 336-337 “.. Moreover, the SCS or sham stimulation was performed for 30 min. The PWTs were measured at 3 and 6 h after the completion of SCS as described above..”
Data are not given. Correct it.
3) Fig.1 a, b shows data obtained in SNI groups before and after SCS or sham stimulation or minocycline administration. However there are no data obtained in sham SNI rats with or without SPS. What is the SCS or minocycline effect in sham SNI rats? Clarify it.
4) As rule, the required level for significance is considered P< 0.05. And, therefore, differences at p<0.09 or p<0.07 are not significant (there is only a trend). It means that SCS-induced effect on pain disappears at 12 h, because there are no significant differences between SCS and sham stimulation. However authors state:
Line 87-88 “..Moreover, PWT at 12 hours after 6 hours of successive SCS was also higher than that obtained in the post-sham stimulation group (p = 0.09) ..”
Line 112-113 “..Moreover, the ipsilateral increased density of Iba1-IR cells on day 3 after SNI with post-sham stimulation was markedly suppressed by i.p. minocycline administration (vehicle (n =3): 9.9 ± 0.9; minocycline (n =3): 6.0 ± 1.3, p = 0.07) (Fig. 3c)…”
Clarify it.
5) How long does SCS-induced analgesic effect last? What is the SCS effect on pain sensitivity and other parameters between 30 min and 12 h.
6) Did electrical stimulation applied to the left forepaw or hindpaw cause spinal reflex in anaesthetized animals?
7) Fig.3 Data are given for Sham group (sham SNI +sham stimulation), but they are absent for group “sham SNI + SCS “.
SCS-induced suppression of microglial activation at 12 h was significant (p<0.01) (Fig. 3) whereas analgesic effect disappeared at 12 h (P< 0.09) (Fig. 1). Discuss it.
Author Response
The manuscript describes an original research performed in rats focusing on the mechanisms underlying spinal cord stimulation (SCS)–induced analgesic effect on neuropathic pain. Authors investigated hypothesis that SCS-induced analgesic effect on neuropathic pain may be due to suppression of the spared nerve injury (SNI)-induced microglial activation in spinal cord. This idea is supported by results from experiments using immunohistochemistry, electrophysiological and optical imaging methods. Beneficial effect of SCS on neuropathic pain is associated with microglial inactivation which is accompanied by suppression of spinal neurons hyperexcitability and reduction of SNI-activated area within somatosensory cortex.
Remarks and comments:
1) Line 57-59 “..At present, the exact mechanisms underlying the analgesic effect attributable to SCS remain poorly understood…”
Has SCS only analgesic effect? Or SCS exerts dual action on pain in humans and animals? Add this information in Introduction or Discussion.
# We have added the sentences in Introduction or Discussion section as follows;
L62-65. “Sciatic nerve injuries have been widely employed as an animal model of neuropathic pain, and SCS reportedly improves the intractable neuropathic pain following nerve injuries [5, 6, 7, 8, 9]. Presently, the prime analgesic effects of SCS involves the activation of the descending pain inhibitory system [4, 10].”
L295-297. “Several studies suggest that SCS induces an analgesic effect on pain hypersensitivity in neuropathic pain models [7, 9, 38, 44], and a major clinical strength of SCS is that it can often dramatically reduce the intensity of neuropathic pain which other therapies have no effect on [45].”
2) Line 89-90 “..On the other hand, SCS for thirty minutes did not recover the decrement of PWT at 3 and 6 hours post-SCS (post-sham stimulation (n = 7), SCS (n = 7)) …”
Line 336-337 “.. Moreover, the SCS or sham stimulation was performed for 30 min. The PWTs were measured at 3 and 6 h after the completion of SCS as described above..”
Data are not given. Correct it.
# We have deleted the above sentences.
3) Fig.1 a, b shows data obtained in SNI groups before and after SCS or sham stimulation or minocycline administration. However there are no data obtained in sham SNI rats with or without SPS. What is the SCS or minocycline effect in sham SNI rats? Clarify it.
# We have not performed sham treatment for SNI. Therefore, there are no data obtained in sham SNI rats with or without SCS and minocycline administration.
4) As rule, the required level for significance is considered P< 0.05. And, therefore, differences at p<0.09 or p<0.07 are not significant (there is only a trend). It means that SCS-induced effect on pain disappears at 12 h, because there are no significant differences between SCS and sham stimulation. However authors state:
Line 87-88 “..Moreover, PWT at 12 hours after 6 hours of successive SCS was also higher than that obtained in the post-sham stimulation group (p = 0.09) ..”
Line 112-113 “..Moreover, the ipsilateral increased density of Iba1-IR cells on day 3 after SNI with post-sham stimulation was markedly suppressed by i.p. minocycline administration (vehicle (n =3): 9.9 ± 0.9; minocycline (n =3): 6.0 ± 1.3, p = 0.07) (Fig. 3c)…”
Clarify it.
# We have modified the sentences as follows;
L95-96. “Moreover, PWT at 12 hours after 6 hours of successive SCS tended to be higher than that obtained in the post-sham stimulation group (p = 0.09).”
L118-120. “Moreover, the ipsilateral increased density of Iba1-IR cells on day 3 after SNI with post-sham stimulation tended to be suppressed by i.p. minocycline administration (vehicle (saline) (n =3): 9.9 ± 0.9; minocycline (n =3): 6.0 ± 1.3, p = 0.07) (Fig. 3c).”
5) How long does SCS-induced analgesic effect last? What is the SCS effect on pain sensitivity and other parameters between 30 min and 12 h.
# Though we have revealed that the decrement of PWTs was recovered at 30 minutes 12 hours (there is only a trend) post-SCS on day 3 after SNI, it is unknown that how long the SCS-induced analgesic effect will last in this study. However, previous study has indicated that SCS for 6h showed significantly greater analgesia in neuropathic pain model, and the analgesic effect disappear on day 1 after SCS for 6h (Anesth Analg. 118(2):464-72. 2014). Consequently, the SCS-induced analgesic effect is expected to disappear within 24 hours.
We have not measured other parameters between 30 min and 12 h in this study. However, visible physiological changes such as abnormal behavior, bleeding, changes in heart rate, respiratory rate, and Body temperature) were not observed.
6) Did electrical stimulation applied to the left forepaw or hindpaw cause spinal reflex in anaesthetized animals?
# We have added the sentences as follows;
L461-462. “As above-mentioned anesthesia with urethane, the electrical stimulation-induced spinal reflex was not observed visually.”
7) Fig.3 Data are given for Sham group (sham SNI +sham stimulation), but they are absent for group “sham SNI + SCS “. SCS-induced suppression of microglial activation at 12 h was significant (p<0.01) (Fig. 3) whereas analgesic effect disappeared at 12 h (P< 0.09) (Fig. 1). Discuss it.
# Actually, we have not performed the analysis of microglial activation in the sham group (sham SNI + SCS). Nevertheless, there was no significant difference in the contralateral (uninjured sciatic nerve side) density of Iba1-IR cells on day 3 after ipsilateral SNI with SCS stimulation. Therefore, it is assumed that SCS does not effect on microglial activity under normal conditions.
At present, the prime analgesic effects of SCS are considered as activation of descending pain inhibitory system (Neurosurg Clin N Am, 30,169-194. 2019; IEEE Trans Biomed Eng. 62, 1604-13. 2015). Even if microglial activity is completely suppressed by SCS at 12 h, the SCS-induced analgesic effect by activation of descending pain inhibitory system may disappear at 12 h. Further research is needed in the future.
We have added the sentences as follows;
L62-65. “Sciatic nerve injuries have been widely employed as an animal model of neuropathic pain, and SCS reportedly improves the intractable neuropathic pain following nerve injuries [5, 6, 7, 8, 9]. Presently, the prime analgesic effects of SCS involves the activation of the descending pain inhibitory system [4, 10].”
Round 2
Reviewer 2 Report
Authors have addressed the comments adequately in response letter with reflected changes in the revised version. There is no further comment.
Author Response

(The authors gave the same response as above.)

Reviewer 3 Report
The authors have carefully replied to all the concerns of the reviewer. Only one point needs additional consideration. Since the authors have not performed sham treatment for SNI, this should be explained or discussed.
Line 107 “..minocycline (30 mg/kg/day) administration (and PWT in naive rats. (Mann–Whitney U test). Delete bracket.
Author Response
The authors have carefully replied to all the concerns of the reviewer. Only one point needs additional consideration. Since the authors have not performed sham treatment for SNI, this should be explained or discussed.
# We have evaluated the analgesic mechanisms of the SCS on neuropathic pain following SNI in this study, not to compare the SCS effect on physiological pain and the SCS effect on SNI-induced neuropathic pain. Therefore, we have not performed sham treatment for SNI.
Line 107 “..minocycline (30 mg/kg/day) administration (and PWT in naive rats. (Mann–Whitney U test). Delete bracket.
# We have deleted the bracket.